# Simulation and Verification of Vertical Heterogeneity Spectral Response of Winter Wheat Based on the mSCOPE Model

**DOI:** 10.3390/s20164570

**Published:** 2020-08-14

**Authors:** Linsheng Huang, Yuanyuan Zhang, Guijun Yang, Dong Liang, Heli Li, Zhenhai Li, Xiaodong Yang

**Affiliations:** 1National Engineering Research Center for Agro-Ecological Big Data Analysis & Application, Anhui University, Hefei 230601, China; linsheng0808@ahu.edu.cn (L.H.); ZhangOo2020@163.com (Y.Z.); dliang@ahu.edu.cn (D.L.); 2Beijing Engineering Research Center of Agricultural Internet of Things, Beijing 100097, China; lihl@nercita.org.cn (H.L.); lizh@nercita.org.cn (Z.L.); yangxd@nercita.org.cn (X.Y.)

**Keywords:** SCOPE, sensitivity analysis, FAST, vertical heterogeneity, vegetation index, spectral reflectance, winter wheat

## Abstract

Vertical heterogeneity of the biochemical characteristics of crop canopy is important in diagnosing and monitoring nutrition, disease, and crop yield via remote sensing. However, the research on vertical isomerism was not comprehensive. Experiments were carried out from the two levels of simulation and verification to analyze the applicability of this recently development model. Effects of winter wheat on spectrum were studied when input different structure parameters (e.g., leaf area index (LAI)) and physicochemical parameters (e.g., chlorophyll content ($Chl_{a+b}$) and water content ($C_{w}$)) to the mSCOPE (Soil Canopy Observation, Photochemistry, and Energy fluxes) model. The maximum operating efficiency was 127.43, when the winter wheat was stratified into three layers. Meanwhile, the simulation results also proved that: the vertical profile of LAI had an influence on canopy reflectance in almost all bands; the vertical profile of $Chl_{a+b}$ mainly affected the reflectivity of visible region; the vertical profile of $C_{w}$ only affected the near-infrared reflectance. The verification results showed that the vegetation indexes (VIs) selected of different bands were strongly correlated with the parameters of the canopy. LAI, $Chl_{a+b}$ and $C_{w}$ affected VIs estimation related to LAI, $Chl_{a+b}$ and $C_{w}$ respectively. The Root Mean Square Error (RMSE) of the new-proposed $NDVI_{green}$ was the smallest, which was 0.05. Sensitivity analysis showed that the spectrum was more sensitive to changes in upper layer parameters, which verified the rationality of mSCOPE model in explaining the law that light penetration in vertical nonuniform canopy gradually decreases with the increase of layers.

## 1. Introduction

Vegetation modelling is a nondestructive technique for quantifying vegetation properties and analysing the physiological conditions of crops, such as growth potential and nutrition [1]. The soil canopy observation, photochemistry, and energy fluxes (SCOPE) model simulates the transmission law of light in uniform canopy [2]. However, the SCOPE model is a vertically integrated radiative transfer and energy balance model based on the classical 1-D SAIL model [3,4]. Disregarding vertical heterogeneity in vegetation canopy may negatively affect fluorescence and reflectivity spectra, a condition that leads to erroneous interpretation of the physiological characteristics of natural plants [4]. In reality, the biochemical components of the canopy with 3D structure are not evenly distributed and exhibit vertical heterogeneity. Therefore, the influence of vertical heterogeneity on spectral response is not negligible [5,6,7]. Furthermore, vertical heterogeneity may greatly influence the accuracy of actual growth detection and remote estimation of nutrient characteristics of crops [8]. Zhao et al. [8] also found that the chlorophyll, leaf moisture, and leaf area index (LAI) of winter wheat have vertical heterogeneity. Moreover, the physiological and biochemistry indices of crops and canopy are correlated with the spectra at visible light, near-infrared, and middle-infrared bands [8,9].

Remote sensing techniques are rapidly developing. These techniques allow time-effective and noninvasive data collection over large scales [10,11,12,13,14]. Remote sensing techniques have two directions in estimating vegetation characteristics: one is an empirical statistical method for analysing the physiological variables and spectral vegetation indexes (VIs) of vegetation; the other involves the establishment of a physical canopy reflection model [15]. Harnessing the advantages of spectral technology will allow the accurate and efficient monitoring and evaluation of the growth and yield of large-scale crops [16]. However, spectral technology can obtain canopy spectra only. Combining the spectral data obtained by satellite and the vegetation model mentioned herein can invert the biochemical information of different layers in vegetation. On the basis of the SCOPE model, Yang et al. [3] developed the mSCOPE model. This model adds vertical stratification of crops to explain better the correlations between remote sensing observations and plant functional traits. However, few studies using this multilayer model have been conducted on the spectral effects of crops on vertical biochemical contents. The interdependence between vegetation biophysical parameters and spectral data must be understood to infer crop status from spectral data that represent the shape of crops and the colour of their leaves [17]. The laws of radiative transfer within vegetation can be obtained through empirical and theoretical studies [17]. In the field, the statistical distribution of reflectivity denotes the comprehensive effects of canopy variables, soil environment, growth stage, and other factors. Numerous studies have approximated this reflectivity as a Gaussian distribution. However, in practice, this assumption is too harsh because the physiological distribution of the canopy is always not uniform. Leaf area index (LAI, $m^{2}leaf/m^{2}\;soil$) [16], chlorophyll ($Chl_{a+b}$, $\mu g/cm^{2}$) [18], and vegetation water content ($C_{w}$, $g/{\mathrm{cm}}^{2}\;\mathrm{fresh}\;\mathrm{weight}$) [19] are important parameters used in characterizing photosynthesis, biomass, and evapotranspiration. Previous studies have demonstrated that many remote sensing spectral vegetation indexes have good correlation with LAI. Shibayama et al. [20] confirmed the correlation between LAI and spectra. The normalized vegetation index (NDVI) is closely related to the biochemical parameters of crop canopy scale [21]. Tan et al. [16] found that the ratio of near-infrared to green light (reflectance at 810 and 560 nm) has a remarkable relationship with LAI. In consideration of the penetration characteristics of chlorophyll absorption feature bands in inner canopy, a combination of sensitive spectral bands (near-infrared band at 810 nm and green band at 560 nm) was selected to evaluate vertically layered chlorophyll [22]. Previous studies that adopted hyperspectral technique in measuring the water content in the upper, middle, and lower layers of potato have revealed different spectral characteristic reflectance and water content distributions in different leaf positions [19]. Previous works have observed good correlations between vegetation indexes and water content. Wang et al. [23] found that the reflectivity ratios of 1450 and 1940 nm have a good effect on water condition estimation. Ceccato et al. [24,25] highlighted that the reflectance ratios of 1600 and 820 nm have a strong correlation with the equivalent water thickness of canopy leaves. Furthermore, sun-induced chlorophyll fluorescence (SIF) can track the photosynthetic activity of vegetation, and this information is important in determining plant stress and yield [26]. The measurement of chlorophyll fluorescence is nondestructive and noninvasive [27]. The effects of SIF on plant physiology can be monitored by exploring the relationship between reflectance and fluorescence spectrum and chlorophyll. The effects of chlorophyll concentration on chlorophyll fluorescence ratio ($F_{685}/F_{730}$) in intact leaves mainly depends on the reabsorption of fluorescence in the 685 nm band, which increases with the increase in chlorophyll concentration [18]. Owing to the complexity of internal and external factors in the canopy [28], the relationship between spectral characteristics of a single band and canopy variables is usually not universal. Converting vegetation index from spectral data, which always contain at least two or three spectral bands to minimize interference from external factors [28], is simple with high computational efficiency, but it lacks universality. Thus, combining empirical methods and models is necessary. Klem et al. [29] emphasized the influence of canopy vertical heterogeneity on spectral reflectivity by calculating the spectral index of spring wheat in different vertical parts. The mSCOPE model proposed by Yang et al. can be used as an effective tool to study the influence of vertical heterogeneity on canopy reflectance. On the basis of the understanding of the law of radiation transmission in the canopy, the authors extended the 1D model SCOPE to vertical stratification and simply verified the performance of the mSCOPE model in simulating canopy reflectivity. However, the model requires further verification to analyse the influence of various physiological parameters. The reliability and generalization of a model must be verified via sensitivity analysis before it is employed. On the basis of a global sensitivity analysis of selected stratified crop parameters via the Fourier amplitude sensitivity test (FAST), we discussed the sensitivity of these selected parameters to the spectrum of winter wheat [30]. FAST is a sensitivity test method that converts all multidimensional integrals of the input of an uncertain model into one-dimensional integrals [31,32]. FAST mainly calculates the variances and means of results by combining the distribution ranges of input factors. The transformation from multidimensional to one-dimensional is essentially the sensitivity analysis of means and variances. First-order and global sensitivity analyses of the selected parameters were conducted to explore the executable ability in this model. Winter wheat, an important crop worldwide, was selected to test this model. The reflectivity and fluorescence spectra of winter wheat at different vertical strata were measured using the mSCOPE model. Furthermore, vegetation index was calculated from simulation data to analyse further the physiological state of vegetation. Sensitivity analysis (SA) helps in selecting the parameter factors that contribute most to the instability of model output. Furthermore, SA simplifies the difficulty of analysis and provides the necessary foundation and prerequisite for the subsequent simplification of model running time.

The purpose of this study is to explore the method of simulating the winter wheat spectrum with the mSCOPE model and verifying its rationality from multiple perspectives. Firstly, the appropriate number of layers was selected. Based on the spectral data generated by mSCOPE simulation, the correlation between vegetation index and spectrum was analysed. Then, the accuracy of vegetation index (VIs) inversion was evaluated using statistical indicators. Finally, SA was carried out to further verify the rationality of model operation and parameter selection. The rest of this study was organized as follows. Section 2.1 introduced the parameterization of mSCOPE model; Section 2.2 introduced the selection of vegetation index; in Section 2.3, two scenarios were designed, which were used for selecting the optimal number of layers and for accuracy assessment using statistical indicators, respectively; in Section 2.4, SA was applied to verify the accuracy. Section 3.1 gave the analysis results between simulated spectral and VIs; in Section 3.2, the inversion accuracy was verified by field measured data; Section 3.3 gave the SA results. In Section 4, the strengths and weaknesses of the developed approach were analysed in detail. Section 5 summarized the results of this study.

## 2. Materials and Methods

### 2.1. Model Parameterization and Calibration

The winter wheat data of Xiaotangshan National Precision Agriculture Research Demonstration Base in 2015 were combined with the biochemical parameters of the PROSPECT model as a reference to identify a reasonable parameter range as inputs (Table 1). According to different vertical distribution scenarios of canopy biochemical parameter contents, the contents of LAI, $Chl_{a+b}$, and $C_{w}$ at different vertical canopy strata were measured. Input indicators under the vertical distribution profiles of crop parameters in different scenes were given. Only the parameters selected herein were changed according to the requirements of the scene and other parameters were not changed with the field measurement simulations. These values were selected after careful consideration and in accordance with actual situations. To measure the influence of specific parameters on model output, we changed the values of research parameters according to different research scenarios and retained the standard values of other parameters. The distribution of LAI, chlorophyll, and water content initially increased and then decreased in the vertical layers. The middle layer had the highest content.

The mSCOPE model was used to obtain spectral data. Vertical stratification was performed on the basis of the SCOPE model. The modules of the mSCOPE model still followed the SCOPE model and mainly included four modules: one biochemical model (Fluspect) and three energy balance routines (RTMo, RTMf, and RTMt). The Fluspect module simulates the reflectivity, transmittance, and fluorescence of a blade [33]. RTMo and RTMf are used to calculate the radiation transmission of incident radiation and emitted fluorescence in canopy, respectively [34]. RTMo, RTMf, and the energy balance model are closely related to each other [4]. First, light is incident on the leaves, and the excitation mechanism inside the leaves is triggered to carry out photosynthesis. Moreover, certain optical phenomena, such as reflection, refraction, and transmission, occur in the canopy. This process can be mathematically summarized as the following algorithm: first, Fluspect is used to calculate the fluorescence excitation–emission matrices as the required input for RTMo and RTMf; second, RTMo predicts the distribution of irradiance and net radiation on surface elements (leaves and soil); finally, the algorithm inputs the elements calculated in the first two steps into RTMt and RTMf. The mSCOPE model uses soil reflectance and total incident crown radiation, which are easily obtained, to calculate surface reflectance layer by layer from the bottom to the top. The vertical flux profile of each layer is then derived from the top to the bottom. According to the main input parameters of the SCOPE model (Table 2), the mSCOPE model adds 60 vertical sublayers to input additional parameters (Table 3), and then any number of N layers less than or equal to 60 can be chosen during simulation. When one of the varying parameters selected the layered value, the standard values were selected for the other parameters. The values for each layer were entered in sequence. The carotenoids were set at 25% of $Chl_{a+b}$ in these experiments.

The mSCOPE model stratifies the canopy in accordance with the principle of uniformity. The height of each layer is 1/N of the total height from the soil to the top of the canopy. In order to simplify the calculation and facilitate the implementation of the algorithm, the average height hierarchical way was carried out. The mSCOPE model integrated the vertical changes of vegetation attributes, but it did not consider the horizontal changes. Therefore, mSCOPE can be regarded as a 2D model (Figure 1). The mSCOPE model maintained the same model structure and output were the same as SCOPE but adopted different solutions for incident emission radiation transmission in vegetation canopy. The reflectivity was solved by bottom-up “adding” method and the flux profile was calculated by top-down “peeling” method. The specific algorithm implementation was shown in Figure 2. In conclusion, stratification can be extended to any multistratification mode between 2–60 layers on the premise of considering both accuracy and cost performance. Considering the labour cost and operation difficulty, field measurement verification data generally adopt three layers according to the highly uniform stratification of winter wheat. The parameter values of each layer were obtained through field measurement. Then they were inputted into the input table provided by mSCOPE, and the program will automatically call to simulate the results. The model simulates the spectra of top of canopy (TOC) reflected radiation, fluorescence emission in the observed direction, and light synthesis of leaf characteristics, vegetation structure, and microscopic meteorological conditions. These spectra help in expanding our understanding of remote sensing data and photosynthetic mechanism of canopy. After successfully running the model in Matlab2017 to obtain experimental data, Excel and Origin were used to analyse the effects of input parameters on spectral characteristics and calculate the correlation of vegetation index analysis. Simlab2.2 was used to analyse the sensitivity of the mSCOPE model.

### 2.2. Selection of Relevant Vegetation Indicators

This study referred to several spectral indexes proposed by predecessors to make a further analysis between the input parameters of mSCOPE model and the spectrum to study the correlation between them. These spectral VIs were mainly divided into two categories: two-band spectral index and three-band spectral index (Table 4). Due to the difficulty in measuring the spectrum inside the canopy, the internal stratified VIs could not be verified. We collectively analysed the influence of different vertical distributions of three main parameters on the canopy vegetation index. Two LAI-related, two chlorophyll-related, and four water-related vegetation indexes were selected to monitor the effects of model inversion. In analysing the correlation between LAI and related VIs, we selected the NDVI of near-infrared light combined with red light bands and $CI_{green}$ of 810 nm combined with 560 nm to calculate the vegetation indexes of canopy. In analysing chlorophyll-sensitive bands, we chose the green chlorophyll index, that is, $CI_{green}$ of 810 nm combined with 560 nm. The original vegetation index $NDVI_{green}$ (810 nm combined with 560 nm) was also used for $Chl_{a+b}$. Water index (WI), two type of Water ratio vegetation index (WRVI), and normalized difference water index (NDWI) were chosen to investigate the correlation between VIs and canopy moisture content with vertical variation.

### 2.3. Scenario Design

#### 2.3.1. Using Seven Synthetic Datasets to Verify the Optimal Number of Winter Wheat Canopy Stratification Number

The canopy was divided into 1 up to 60 layers in the mSCOPE model. To select the optimal number of layers, we conducted seven experiments, in which LAI and chlorophyll content were selected as the controlled variables. Because water content mainly affects the near-infrared band and has little influence on the vegetation index selected to evaluate the results, it can be ignored in this scenario. The experimental scenario was set as shown in Table 5. The parameter distribution of different levels from the top to the bottom of the canopy was shown from left to right as enumerated in Table 5. When the stratification number was i, LAI and $Chl_{a+b}$ selected the values of the corresponding i-th layer, respectively. The number of parameters entered into the model represent the number of layers. In this experiment, we selected standard reference values for the other parameters as input listed in Table 2. For example, water content was 0.009 cm.

#### 2.3.2. Statistical Indicators to Evaluate the Accuracy of 21 Real Scenes

In order to distinguish the simulation scene from that of the simulation work, the latest winter wheat data of Xiaotangshan National Precision Agriculture Research and Demonstration Base in April 2020 was selected to verify the inversion accuracy of the model. The parameters in Table 6 were changed according to different scenes, and other parameters were retained in the field measurement simulation. The units of equivalent water thickness of chlorophyll leaves and LAI are shown in Table 5, which will not be repeated later. In the experiment, the real data measured in the field was input into the mSCOPE model. The standard input parameters inherent in the model were selected as additional parameters without changing them during the routine.

The simulated spectrum was obtained and then the corresponding VIs were calculated. In order to analyse the accuracy of simulation results of mSCOPE model, we compared simulated VIs values with real VIs values of 21 groups through deviation (Bias) and root mean square error (RMSE) analysis. The formula for calculating deviations between measured and simulated values is as follows:(1)$$Bias=\frac{\hat{y}\;-y}{\hat{y}}.$$

We also selected RMSE to evaluate systematically the overall accuracy of the 21 group inversions. The formula is
(2)$$\mathrm{RMSE}\;=\;\sqrt{\frac{1}{N}\sum_{i=1}^{N}{\left({\hat{y}}_{i}\;-y_{i}\right)}^{2}},$$
where *y* and $\hat{y}$ represent the measured and simulated VIs of different scenes, respectively; *N* indicates the total experiment numbers; and *i* represents the i-th experiment. In the experiment, 21 different scenarios from dense to sparse without removing ear of winter wheat in booting stage were selected for simulation (Figure 3). The number of stems sampled is usually 50 single stems. Leaves in the upper-layer are all green leaves. the middle-layer leaves are treated individually with a small amount of unseparated yellow leaves and the lower-layer leaves are divided into green leaves and yellow leaves.

### 2.4. Data Generation and Processing of Sensitivity Analysis

FAST is a sensitivity test method, which is mainly used to calculate the variance and mean value of simulation results by combining the distribution range of input factors. The experimental steps are as follows (Figure 4). Firstly, the parameters need to be sampled before SA. Simlab2.2 provides various sampling algorithm choices for different sample sets, and the selection of sampling methods affects the type of sensitivity analysis in the later stage. In this study, the FAST method was used to conduct 585 experiments to meet the accuracy requirements of the method. The upper and lower limits of the selected parameter factors are shown in Table 7. The input parameters are then programmed to loop into mSCOPE and the output is saved in a TXT format document in the format specified by Simlab. Finally, we selected the TXT file under the Simlab interface, performed Monte Carlo simulation, and sensitivity analysis was conducted through the postprocessor.

The FAST method applies to both monotone and nonmonotone models. This method can not only perform first-order sensitivity analysis but also obtain global sensitivity analysis results, which makes up for the fuzziness of global influence of single variable. Furthermore, this method assumes a relationship between inputs and outputs. The principle of FAST analysis method can be divided into four stages. First, in the space defined by the input parameters, the expected value and variance of *y* were expressed in the form of an integral, and then they were estimated. Second, multidimensional integral transformation was defined as a one-dimensional integral. Third, the expected value and variance of *y* were then estimated. Finally, the sensitivity index of *y* was calculated. The first-order sensitivity index and the total sensitivity index were obtained by rapidly expanding the same set of models for n times by using the terms in the Fourier decomposition of the model output. The number *i* of LAIi, $Chl_{a+b}i$ and $C_{w}i$ in Table 7 represent the upper, middle, and lower layers of the canopy respectively.

## 3. Results

### 3.1. Using mSCOPE Model to Simulate Spectrum of Winter Wheat

The relationship between vegetation biochemical content and spectral reflectance characteristics could be easily investigated by changing a series of parameters in the mSCOPE model. That is hard to realize in real field experiments. Previous studies found that influencing factors of the spectrum from visible light to near-infrared bands could be described as follows [38,39,40,41]: the absorption segment of visible light from 500–750 nm was mainly affected by pigments (such as chlorophylls a and b, carotene and xanthophylls); the near-infrared (SWNI) band of 750–1350 nm was greatly affected by the internal structure of the blade; the medium and long wave infrared bands (MWNI and LWNI) of 1350–2500 nm were mainly affected by the water content in tissues. Spectral analysis of different LAI, $Chl_{a+b}$ and water content were conducted to analyze and verify the interpretability of layered spectral analysis by mSCOPE model. Furthermore, the first derivative of the spectral data was analysed. Derivative spectroscopy can compress the influence of background noise and low-frequency signal, and it is helpful to eliminate the influence of low-frequency spectral components such as soil background and atmosphere on the target.

#### 3.1.1. Selection of the Optimal Stratification Number for Winter Wheat Canopy in mSCOPE Model

To evaluate the adaptability of the mSCOPE model in simulating the relationship between spectra and vegetation biochemical contents, we selected the three major agronomic parameters LAI, $Chl_{a+b}$, and $C_{w}$ as stratified simulation variables. The number of canopy layers selected was three. Through seven experiments, we found that the efficiency divided into three layers was higher and the calculation time could be reduced and ensure the accuracy. Figure 5 provides a schematic of NDVI analysis for one to seven layers. With regard to simulation time consumption, the simulation time of four layers was about 1.8 times longer than that of three layers. Moreover, their accuracy was not considerably different, and the accuracy of three layers was enough. The higher the layers, the slower the simulation speed would be. Furthermore, the accuracy did not substantially improve. The mean values of the 7 stratified NDVI was calculated. We use the average value to minus the measure value and the result divided by the average value. Thus, the error rate (ER) was computed in Equation (3):(3)$$ER_{i}=\frac{\hat{y}\;-y_{i}}{\hat{y}},$$
$\hat{y}$ was the average of these groups. $ER_{i}$ and $y_{i}$ represented ER and the NDVI value of the i-th experiment respectively. The accuracy of the five layered canopy is at the expense of time. The computing time increases 2.3 times over from three layers to five layers. Considering time and accuracy, three-layers simulation was a more reasonable choice. To further verify the conclusion, we calculated the operating efficiency by time and error, and the formula is as follows:(4)$$E=\frac{1}{T*A}$$

The program run time (*T*) which was the running time when output the layered spectrum, was measured in seconds. The error (*A*) was obtained from the absolute difference between the simulated vegetation index and the mean value. Efficiency is inversely proportional to time (*T*) and error (*A*). The statistics of running time error and efficiency are shown in Table 8.

Note that in order to balance the weight ratio of time and error, we multiply the time with the weight of 60 to convert it into minutes. Figure 5 shows the simulation efficiency of the 7 layers drawn according to Table 8. It could be seen from Table 8 and Figure 5 that the efficiency of the three layers is the highest. The operation efficiency in our experiment is an inverse proportional function of time and error. When the number of layers rises, calculation amount and time consumption increase, while the accuracy rises to a limited extent. Three, four, and five layers are better options when data acquisition conditions allow. However, the high number of layers will make it more difficult to collect data, and the cost of labour and material resources is not cost-effective compared with the improved accuracy. Combined with the above analysis on the operating efficiency of the model, it is suggested to choose three layers as the best layer.

#### 3.1.2. Effects of Changing Simulation Parameters on Reflectivity Spectrum

After a relatively reasonable stratification standard was obtained, simulation work goes on smoothly. We made a qualitative analysis of the spectrum obtained from the experiment. Firstly, the influence of the vertical profile of LAI was analysed. Vegetation coverage related to LAI and an increase in vegetation coverage will increase leaf area. Hence, the higher the slope of the red border, the better the vegetation will grow. Although the total LAI content gradually increased, the difference in overall stratified reflectance was marginal, showing only differences in amplitudes. In general, two turning points appeared in the reflectance spectrum in the red and infrared bands. These turning points were mainly determined by the physiological and structural characteristics of the leaves (Figure 6). Figure 6a was the three-layered canopy reflectance diagram of the fifth group which was the standard reference scenario. The upper, middle, and lower layers were made for the winter wheat diagram on the right. $TopR_{i};\;MiddleR_{i};\;LowerR_{i}$ in Figure 6a–c represented the upper, middle, and lower reflectance curves of the i-th experimental. As shown in Figure 6b–d, the upper layer had a lower reflectance in the visible light bands and better light absorption than the other layers. In the experiments, the design of the middle layer LAI value was the largest among the three layers of canopy. When light is transmitted in the canopy, light reflection and absorption by the upper blade will attenuate light radiation that is stronger than the influence of enlarged leaf area of middle layer within a reasonable range. Near the 740 nm band, light absorption by the pigments gradually decreased, whereas light reflection by cells increased. The reflectivity in the near-infrared region is the most sensitive to canopy and blade structure. Water content of the leaves has the most remarkable influence on light reflectivity. An obvious extreme value point of the first derivative near 740 nm appeared (Figure 6). This extreme value point was the inflection point of the reflectance spectrum. When chlorophyll amount was the same, the reflectivity of the upper, middle, and bottom layers followed the same trend. However, the reflectance amplitude was greatly affected by the vertical structure of the canopy, thereby demonstrating the phenomenon that the reflectance of the upper layer was lower but the reflectance of the bottom layer was higher compared with that of the other layers. LAI had a negative correlation with the reflectivity of visible light band, but LAI was remarkably positively correlated with the reflectance of near-infrared band, in which 810 nm had the best correlation.

Next, the effect of chlorophyll vertical profile on spectral characteristics was analysed. As indicated by the three-layered reflectance spectral profile shown in Figure 7, chlorophyll mainly affected the visible light band. Figure 7a is the three-layered canopy reflectance diagram of the eighth group which was the standard reference scenario. The upper, middle, and lower layers were made for the winter wheat diagram on the right. $TopR_{i};\;MiddleR_{i};\;LowerR_{i}$ in Figure 7a–c represents the upper, middle, and lower reflectance curves of the i-th experimental group. A green peak appeared in the same position of the three strata at about 550 nm in the green light wave band. The correlation coefficient between $Chl_{a+b}$ and VIs was also strongly correlated with the green light band. An obvious extreme value point of the first derivative appeared near 550 and 710 nm. This point was the inflection point of the reflectance spectrum, indicating that the spectral characteristics changed and the correlation between the spectrum and chlorophyll content was demonstrated. Chlorophyll in cells is in the hydrosol state and has strong infrared reflection. Mesophyll cells contain numerous chloroplasts, which are the main sites of photosynthesis. The reflectance spectra of spongy tissues with large amounts of chlorophyll in the cavity varies greatly.

Finally, the effect of equivalent water thickness on spectral reflectance was analysed (Figure 8). Figure 8a was the three-layered canopy reflectance diagram of the third group which was the standard reference scenario. The upper, middle, and lower layers were made for the winter wheat diagram on the right. $TopR_{i};\;MiddleR_{i};\;LowerR_{i}$ in Figure 8a–c represents the upper, middle, and lower reflectance curves of the i-th experimental group. The water and leaf structures substantially changed because of the reflection characteristics of cells in the leaves after filling with water. The water content of the leaves almost had no effect on the spectral reflectance of visible band because water reflected almost all of the visible light. Results showed that water and leaf structure had a strong influence on the 800 nm band of the near-infrared region, and the reflectance of the top layer was higher than that of the other layers. Three obvious extremum points of the first derivative appeared in the reflectance spectrum from 800 nm to 2400 nm. These points were the characteristic points of concurrent spectral changes. Correlation analysis also verified this result. The selected infrared bands of 820, 1450, 1600, and 1940 nm were also near the spectral change feature points and could well realize the inversion of moisture. Given that the reflection characteristics of the cells in the leaves considerably changed after filling with water, water and leaf structures affected the various characteristics of reflectance spectrum. Water content almost had no effect on the visible band of spectral reflectance because water reflected almost all of the visible light. Results showed that water and leaf structure had a great influence in the 800 nm band of the near-infrared region, and the reflectance of the top layer was higher than that of the other layers.

#### 3.1.3. Effects of Changing Simulation Parameters on Fluorescence Spectra

To verify the simulation effects of the model on fluorescence spectrum, we performed a series of spectral outputs and analysed the influence of the parameters on fluorescence spectrum. SIF is a fluorescence reaction caused by natural light (from 400 nm to 760 nm) during photosynthesis of crops (Figure 9). Compared with the reflectance spectrum, the fluorescence spectrum was directly related to the photosynthetic mechanism and revealed the photosynthetic physiological state of vegetation. Compared with the reflectance spectrum, the acquisition of SIF via nondestructive observation of large areas of vegetation can reveal the physiological state of vegetation during photosynthesis because fluorescence spectrum is directly related to the mechanism of photosynthesis. In this study, the fluorescence spectra of the changing LAI, chlorophyll content, and equivalent water thickness were constructed using the fluorescence data output from the mSCOPE model. Moreover, the relevant fluorescence vegetation index ${\mathrm{RVI}}_{f}$ (fluorescence spectrum ratio of 685–730 nm) was calculated. The $R^{2}$ of LAI, $Chl_{a+b}$, and $C_{w}$ was 0.9707, 0.7662, and 0.9884, respectively. Good correlations of the results indicated that the selected fluorescence index was universal and that the mSCOPE model was robust. The fluorescence gradually decreased as LAI increased (Figure 9a). As chlorophyll concentration increased, the fluorescence increased in the beginning and then weakened thereafter. The fluorescence peak shifted to the right. Compared with the reflectance spectrum, slope gain and right-shift were observed on the red edge, indicating that the fluorescence spectrum could reflect the growth state of crop (Figure 9b). The increase in water content in the leaves would lead to a decrease in fluorescence. Thus, water content and fluorescence had a negative correlation. A fluorescence peak also appeared at 740 nm (Figure 9c). The red edge, which is an important indicator of plant pigment status and health, is closely related to the physical and chemical parameters of vegetation. Therefore, the red edge is an ideal tool for surveying vegetation status via remote sensing. $TOCf_{i}$ represents the canopy fluorescence spectra of the i-th experiment.

#### 3.1.4. Correlation Analysis between Spectral Vegetation Indexes and the Three Selected Parameters

The relationship between vegetation index and biochemical properties of vegetation in specific spectral combinations of bands is also of great interest to us. Firstly, correlation analysis between VIs and LAI was carried out. Since the sensitive bands and robust vegetation index proved in previous experiments were selected, the results obtained were satisfactory. The correlation coefficients of TOC LAI with NDVI and RVI were 0.9687 and 0.9925, respectively, by which the selected VIs showed a good positive correlation between LAI (Table 9). The ratio vegetation index showed a better correlation than that of NDVI. The correlation coefficient between chlorophyll fluorescence index and LAI was 0.9707. Vegetation spectrum was a complex and mixed response of vegetation to a variety of factors such as the shadow of soil environment and atmosphere, which is influenced by many natural factors. The values of different vegetation indices have certain differences and uncertainties, and the effects of some known factors can only be removed by adding and subtracting ratios of different bands of the spectrum. RVI, which was also known as greenness, is defined as the ratio of the reflectivity of two channels, which can better reflect the difference of vegetation coverage and growth status. Therefore, when LAI changes were only controlled in the simulation, the correlation between LAI and RVI was the highest. Then, the analysis of the effect of Cab content on VIs was conducted. The correlation coefficients of $Chl_{a+b}$ related vegetation indexes in TOC with ${\mathrm{CI}}_{\mathrm{green}}$ and ${\mathrm{NDVI}}_{\mathrm{green}}$ were 0.9996 and 0.9917, respectively. ${\mathrm{CI}}_{\mathrm{green}}$ had a slight advantage over ${\mathrm{NDVI}}_{\mathrm{green}}$. The correlation coefficient between chlorophyll fluorescence index and $Chl_{a+b}$ was 0.7662. Finally, the response of vegetation index to water change was analysed. Results of $C_{w}$ related vegetation indexes were calculated as follows: WI was 0.9999; NDWI was 0.9931; the combination between the two bands of 1600 nm and 820 nm with RVI was 0.9918, and that between the two bands of 1450 nm and 1940 nm was 0.9659. WI had the best correlation in the water-related vegetation index. The above analysis results were satisfactory and indicate that vegetation indexes of selected bands had relatively good correlations with the corresponding vegetation parameters. The empirical relationship model was established to calculate the vegetation index from the simulated reflectance spectrum and to establish a statistical relationship with the measured parameters. This method can provide a convenient and efficient reference value for estimating vegetation parameters from the vegetation index extracted for hyperspectral images.

### 3.2. Verification of mSCOPE Performance for Winter Wheat Vegetation Index Inversion

First of all, the real parameters were inputted into the mSCOPE model to obtain the simulated spectrum and the simulated value of VIs. The measured values of VIs were calculated using the measured spectrum, and deviations were calculated together with the simulated value of VIs. We ran 21 sets of real data and calculated the RMSE for each VIs and its maximum and minimum values (Table 10).

The results showed that RMSE of ${\mathrm{NDVI}}_{\mathrm{green}}$ was the smallest, with a value of 0.05. The RMSE of ${\mathrm{CI}}_{\mathrm{green}}$ was relatively large, reaching 1.28, mainly because it was largely influenced by chlorophyll and LAI. It can be seen from Figure 3 that in some cases, LAI selected in the experiment was very small, which will lead to a larger result error. The minimum error values for each VIs in Table 9 also showed small simulation errors and good results under nonextreme experimental conditions. Figure 10 is a schematic drawing of the maximum and minimum values of RMSE and error square, which more intuitively reflects the above conclusions. WRVI1 and WRVI2 represent ${\mathrm{WRVI}}_{\left(1600\mathrm{nm},\;820\mathrm{nm}\right)}$ and ${\mathrm{WRVI}}_{\left(1450\mathrm{nm},\;1940\mathrm{nm}\right)}$, respectively. Because the water conditions in these twelve scenes were all at a reasonable level (equivalent water thickness was between 0.007 and 0.009cm), the error of vegetation index related to water was very small. The decrease of chlorophyll and LAI in winter wheat would cause great fluctuations in ${\mathrm{CI}}_{\mathrm{green}}$, with poor accuracy. The new vegetation index proposed in this paper was named ${\mathrm{NDVI}}_{\mathrm{green}}$ (combination of 810 nm and 560 nm), which is characterized by simple calculation and good robustness. Compared with other vegetation indexes, ${\mathrm{NDVI}}_{\mathrm{green}}$ has a good effect in this experiment. The maximum deviation is kept at a relatively low level and it is not sensitive to the change of physicochemical content in the growth stage of winter wheat. Verification of its universality will be attempted through more experiments in subsequent studies.

### 3.3. SA Results Obtained through FAST

Due to the limited manpower and material resources, it is difficult to obtain and calibrate all the parameters involved in the model. To reduce the number of parameters, SA should be carried out according to local crop varieties and its growth conditions. On the one hand, SA should be performed before a model is used because the extraction of major parameters improves the accuracy and efficiency of simulation work; On the other hand, because the physiological characteristics of different crops (such as LAI and plant height) are generally different, the sensitivity of corresponding parameters should be analysed again to judge if the parameters need to be localized and regionalized.

As can be seen in Figure 11, the difference between the bottom chlorophyll content and other parameters in the first-order SA was very small, resulting in their values clustered together, which cannot be distinguished. However, this problem can be solved easily in global sensitivity analysis, because global SA highlighted subtle differences which can well highlight the variance ratios of weaker parameters. Qualitative analysis from Figure 1 shows that: LAI1, Cw1, and Cab3 all showed higher variance interpretation ability in green light bands; the variance interpretation ability of LAI1 and Cab3 in the red spectral band was optimal, while the influence of other parameters was not obvious; in the near-infrared band, the variance proportion of LAI1, Cw1, and Cab3 gradually decreases. Results of SA showed that the overall trend of first-order and global sensitivities were only locally different, and the overall trend was not different (Figure 12). Nevertheless, the overall sensitivity was more prominent than the first-order sensitivity for the specific expression of parameters. Both the first-order and global sensitivities of the upper-layer LAI were superior in the upper layer of the green light band and the sensitivity indexes reached 0.3507 (one order) and 0.7418 (total order). Specific results were listed in Table 11 and Table 12. The sensitivity index of equivalent water thickness was 0.1964 (one order) and 0.3074 (total order). The global sensitivity index (0.0234) was more sensitive than the single variable-related first-order sensitivity (0.190) of $Chl_{a+b}$. Global sensitivity analysis can extract the relatively important factors and prevent imperceptible avoidance of factors that are not sensitive to a single variable analysis, such as chlorophyll content in the bottom layer in this study. In the sensitivity analysis of red spectral band, the contribution of variance and mean of LAI of the upper layer far exceeded that of the other factors. The sensitivity index was 0.5794 (one order) and 0.8846 (total order). In the first-order sensitivity analysis, the sensitivity index of LAI of the middle layer (0.0593) ranked second. However, in the global sensitivity analysis, the sensitivity of the other parameters showed subtle differences. The sensitivity of the bottom layer $Chl_{a+b}$ (0.1619) was slightly higher than that of the middle layer LAI (0.0961). In the near-infrared band, the water content in the upper layer achieved the dominant position. The sensitivity index of water content was 0.6692 (one order) and 0.9058 (total order), which far exceeded that of the other parameters. The near-infrared band is mainly affected by blade structure. In this band, water is the most important factor.

## 4. Discussion

### 4.1. Canopy Reflectance Modeling of Winter Wheat by Using the mSCOPE Model

Differences in the optical properties of crop canopies were rarely considered. The relationship between vegetation biochemical contents stratified within the canopy and spectral reflectance characteristics could be easily investigated by changing several parameters in the model. However, this step was difficult to realize in actual field experiments. Results indicated that the spectral characteristics changed, and the correlation between the spectrum and chlorophyll content was demonstrated well. The input data of the mSCOPE model could be improved by simplifying its complexity; some model parameters were complicated and difficult to obtain. The initialization values that come with the model may lead to inevitable system errors. Moreover, the inversion process involving inputting different vertical layer parameters was uncertain. The same spectrum may be obtained by adjusting the input variables, and the underdetermined results in practical applications must be judged on the basis of experience to a certain extent. Nevertheless, the mechanism and principle of the mSCOPE model were easy to understand, and its implementation is clear. It had good explanatory ability in simulating photosynthesis and light transmission within the canopy. Moreover, it extended the longitudinal scale of the canopy, and the simulation results it obtains are reasonable, efficient, and accurate. The way the model works can be extended and applied to other homogeneous canopy models. Nitrogen and chlorophyll are closely related. We simply verified that winter wheat with different nitrogen content had a good performance, indicating that this model had a broad application prospect. This conclusion can also be extended to other fields where practical scenarios are combined. For example, diseases and insect pests may cause yellower leaves in the middle layer. Common spectral observation method generally focuses on the top of the canopy. However, this way may cause trouble. When the top layer of leaves can detect spectral changes deterioration, the disease has intensified. Using mSCOPE model to invert physiological indexes such as VIs of canopy interior stratification is undoubtedly a good way to solve the difficulty of internal spectral measurement.

### 4.2. Suitability of the mSCOPE Model in Inversing Vegetation Indexes

The mSCOPE model was used to simulate the spectrum by inputting easily obtained vegetation parameters. The indexes that could not be readily measured were obtained via simple calculation and inversion. In terms of deviations, the simulation effects of the model on NDVI were considerably better than those of RVI. However, the difference in RMSE between the two was not significant. This conclusion requires further verification with more experimental data. Unlike the unstratified inversion accuracy, the measured values of the top layers were used as input parameters of the single-layer canopy. Results showed that its RMSE was greater than 10%, which was substantially higher than stratification accuracy. In the subsequent data collection, as many data as possible required by the model should be collected to eliminate systematic errors and maintain simulation conditions as close as possible to measurement conditions. For parameters that are difficult to obtain, the inversion accuracy and efficiency can be improved by setting a reasonable value range combined with prior knowledge. In general, the fitness of inversion was relatively good, and the simulated value had certain reference value. The best solution to avoid systematic errors is to obtain actual field data. In doing so, model accuracy can be clearly debugged and make adequate preparation for practical applications. The function of this model, which can simultaneously output reflectance and fluorescence spectra, is very powerful. However, this model has many features that must be improved. For example, outputs and inputs are manually performed. A visual interface can be added to save time in running the model and improve its efficiency.

### 4.3. Sensitivity Analysis of Canopy Reflectance to Vertical Profiles of Crop Parameters

The influence of a single-layered variable on the reflectivity of different bands was analysed via the FAST algorithm. Global sensitivity was also obtained. The coupling effects of input parameters were also considered. Given that the physiological characteristics of winter wheat are systematic responses to physicochemical contents, a single aspect is insufficient to explain its biological characteristics. The results also demonstrated that the FAST method can simultaneously analyses the direct and indirect effects and the local and global coupling effects of each parameter. However, the present study provided a simple verification only. Owing to space limitation, we did not compare this method with other sensitivity analysis methods. A follow-up study may be conducted to include this aspect. Nevertheless, the present study has a certain reference value. This method has broad application prospects and may be applicable not only to the mSCOPE model developed herein but also to any other model.

## 5. Conclusions

We outputted the layered reflectance spectra and discussed the advantages and generalization ability of the improved mSCOPE model based on the SCOPE model. Sensitivity indexes of the main parameters were calculated via sensitivity analyses of 585 experiments. Results confirmed that weight ratio was directly related to the advantages and disadvantages of the sensitive factors in spectral reflectance. The effects of vertical distribution of leaf physiological characteristics on reflectance spectra were considered. Scenario simulation analysis revealed that the model was optimal in the case of three layers. Moreover, the vertical distribution patterns of LAI, $Chl_{a+b}$, and $C_{w}$ had an impact on almost all layers of canopy reflectance. LAI had a negative correlation with the reflectance of visible light band and a substantial positive correlation with the reflectance of near-infrared band. Among these bands, 810 nm had the best correlation. Chlorophyll mainly affected visible light bands, and a green peak appeared at about 550 nm in the green light band of which the first derivative was 0. The equivalent water thickness of the blade had almost no effect on visible light but mainly affected the near-infrared bands. In general, incident light radiation is transmitted and reflected by canopy leaves. As light further penetrates down to the canopy, the attenuation degree becomes greater. Therefore, the reflectance spectrum of the upper leaves was more sensitive to changes in the content of vegetation components than that of the lower leaves. The vegetation indexes selected, such as NDVI, RVI, ${\mathrm{CI}}_{\mathrm{green}}$, and WI, were highly sensitive to changes in the corresponding LAI, $Chl_{a+b}$, and $C_{w}$. RVI had a better correlation with LAI than NDVI. ${\mathrm{CI}}_{\mathrm{green}}$ was highly correlated with $Chl_{a+b}$ because this index has a strong reflection attribute to green light. Vertical variations in $C_{w}$ mainly affected Vis related to water attribute estimates, such as WI and NDWI. The simulation results showed that: the operating efficiency of winter wheat canopy stratification was higher than other stratification when it was divided into three layers; LAI vertical profile had an influence on canopy reflectance in almost all bands; the vertical profile of $Chl_{a+b}$ mainly affected the reflectivity of visible region; The vertical profile of $C_{w}$ only affected the near-infrared reflectance. The results showed that there was a strong correlation between the selection of vegetation indexes at different bands and canopy parameters. LAI, $Chl_{a+b}$ and $C_{w}$ affect vegetation index estimation related to LAI $Chl_{a+b}$ and $C_{w}$ respectively, and the RMSE of the proposed new vegetation index ${\mathrm{NDVI}}_{\mathrm{green}}$ was the smallest, which was 0.042. SA showed that the reflectance spectrum was more sensitive to the upper layer parameters. The above experimental results showed that the model had good interpretability and broad application prospects. In the future, we will collect additional experimental data to test the practical applications of the model. We will also apply these data to other models, such as the PROSPECT model, for improvement.

## Figures and Tables

**Figure 1 sensors-20-04570-f001:**
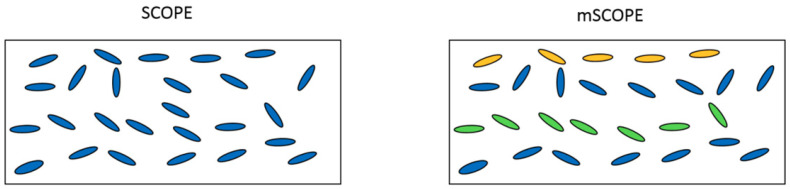
Classification and comparison graphs of different vertical structures of canopy in soil canopy observation, photochemistry, and energy fluxes (SCOPE) and mSCOPE.

**Figure 2 sensors-20-04570-f002:**
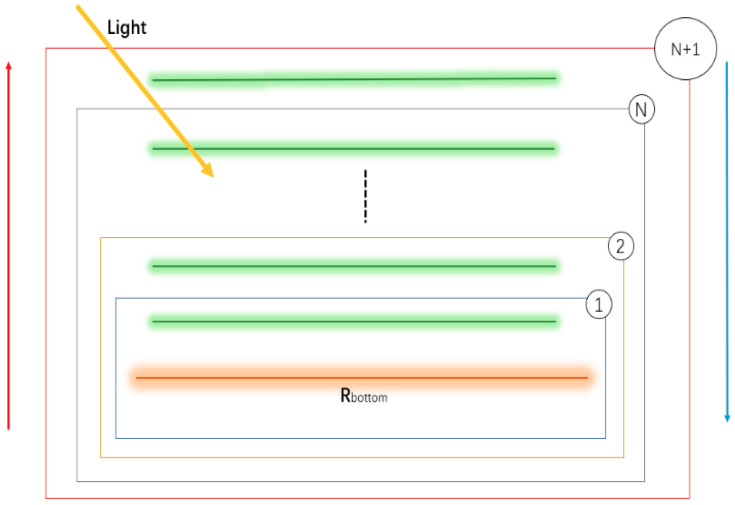
Algorithm implementation diagram of the mSCOPE model.

**Figure 3 sensors-20-04570-f003:**
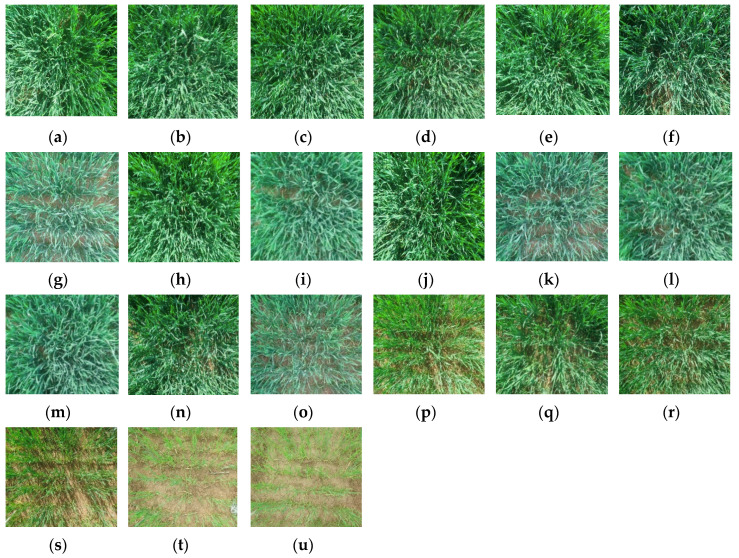
Field plant profiles of winter wheat with different crop physiological contents without removing the ears. The images of 21 plots selected in the experiment were respectively corresponding to (**a**–**u**), and their LAI were gradually increase from 0.78 to 5.33.

**Figure 4 sensors-20-04570-f004:**
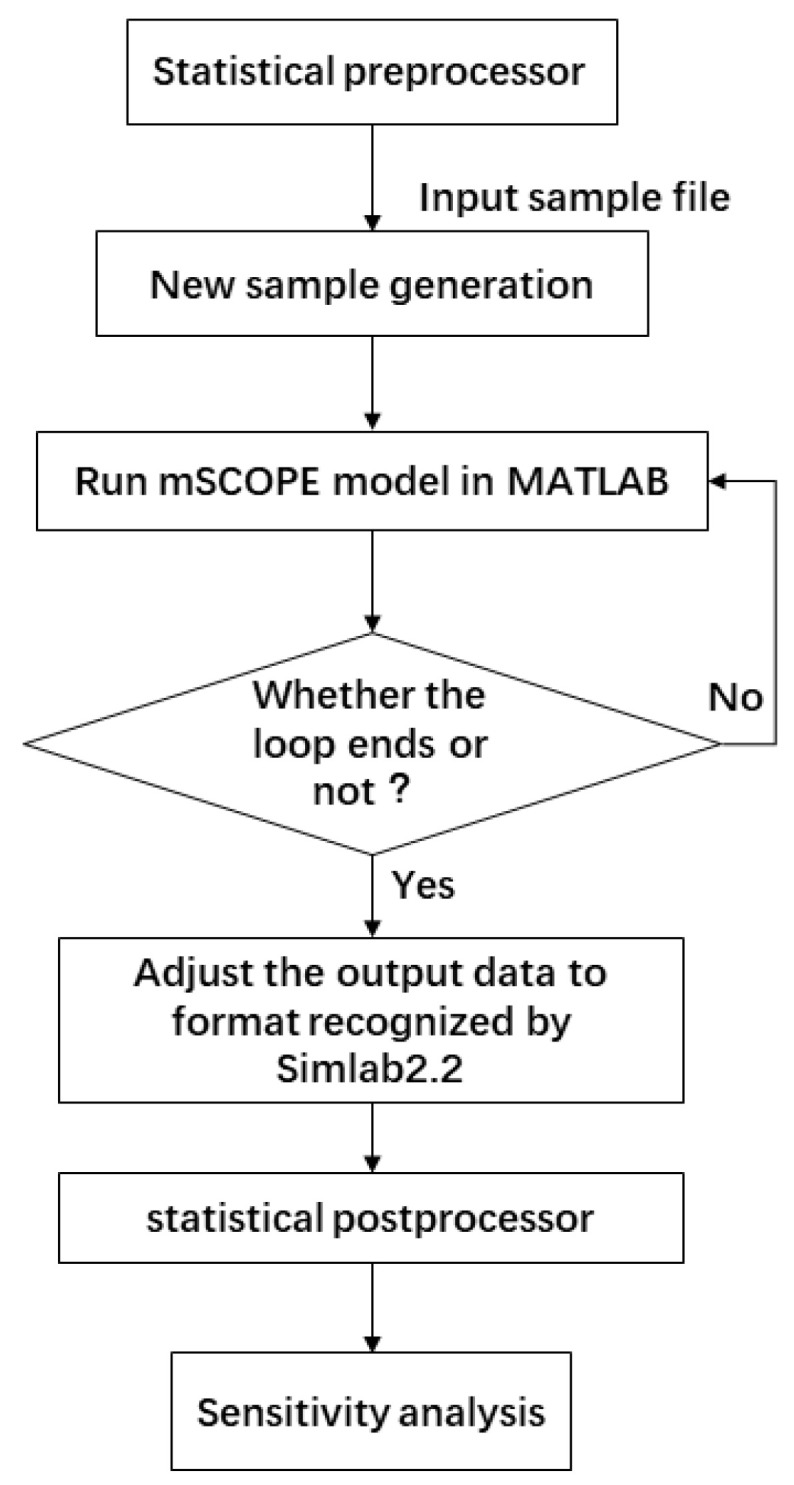
Flow chart of sensitivity analysis.

**Figure 5 sensors-20-04570-f005:**
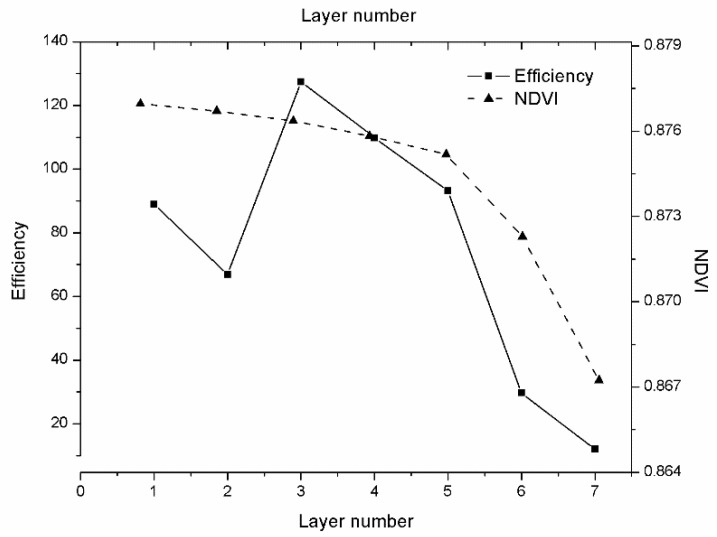
Schematic of normalized vegetation index (NDVI) and Efficiency (E) under seven different layers.

**Figure 6 sensors-20-04570-f006:**
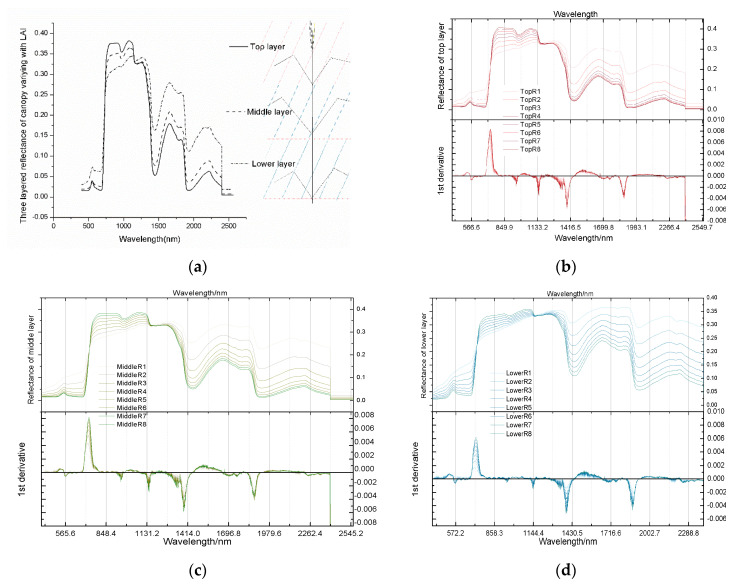
Response diagram of stratified spectra to LAI changes. (**a**) The three-layered canopy reflectance diagram of the fifth group which was the standard reference scenario. (**b**–**d**) represent reflectance spectra of three layers (top, middle, and bottom layers) to vertical changes in LAI and their first derivatives.

**Figure 7 sensors-20-04570-f007:**
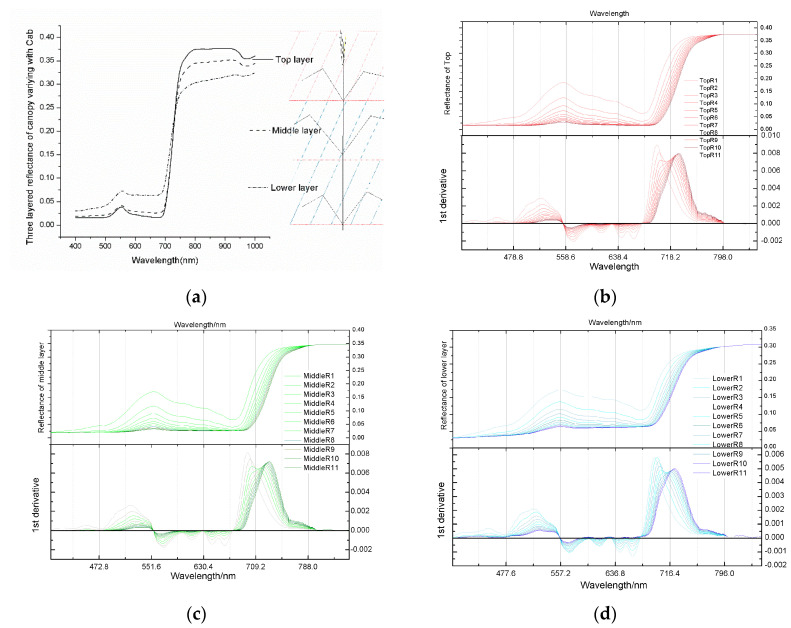
Response diagram of stratified spectra to $Chl_{a+b}$ changes. (**a**) The three-layered canopy reflectance diagram of the eighth group which was the standard reference scenario. (**b**–**d**) represent the reflectance spectra of three layers (top, middle, and bottom layers) to vertical changes in $Chl_{a+b}$ and their first derivatives.

**Figure 8 sensors-20-04570-f008:**
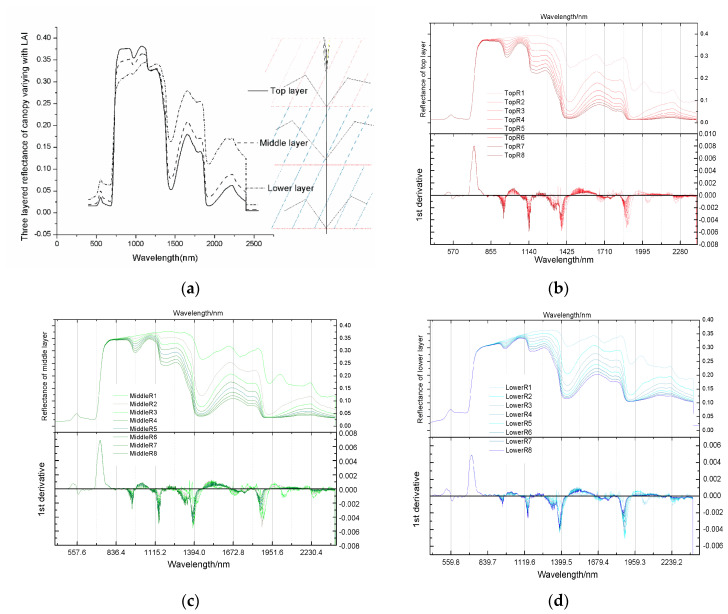
Response diagram of stratified spectra to $C_{w}$ changes. (**a**) The three-layered canopy reflectance diagram of the third group which was the standard reference scenario. (**b**–**d**) represent the reflectance spectra of three layers (top, middle, and bottom layers) to vertical changes in $C_{w}$ and their first derivatives.

**Figure 9 sensors-20-04570-f009:**
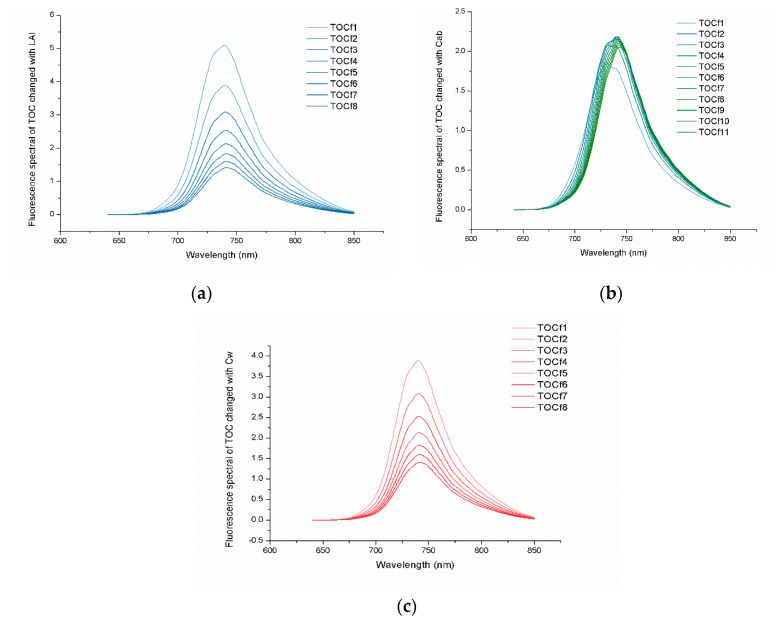
Fluorescence spectra of top of canopy (TOC). (**a**–**c**) represent the fluorescence spectra of TOC changing by LAI, $Chl_{a+b}$, and $C_{w}$ under different scenes. Note: $TOCf_{i}$ in Figure 9 represented the fluorescence spectra curve of the i-th experimental.

**Figure 10 sensors-20-04570-f010:**
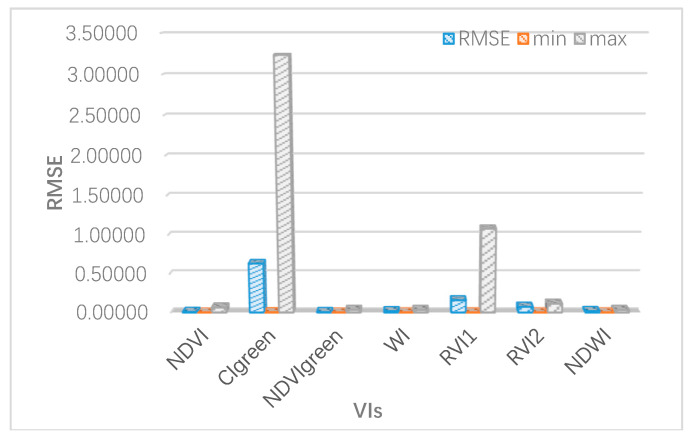
The maximum and minimum values of RMSE, when using the three-tier model to verify accuracy.

**Figure 11 sensors-20-04570-f011:**
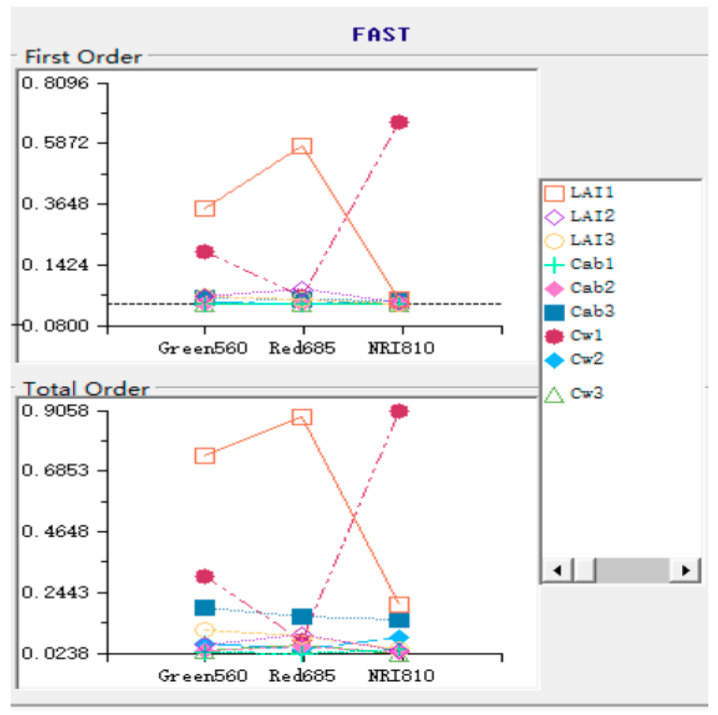
Diagram of first-order and global SA.

**Figure 12 sensors-20-04570-f012:**
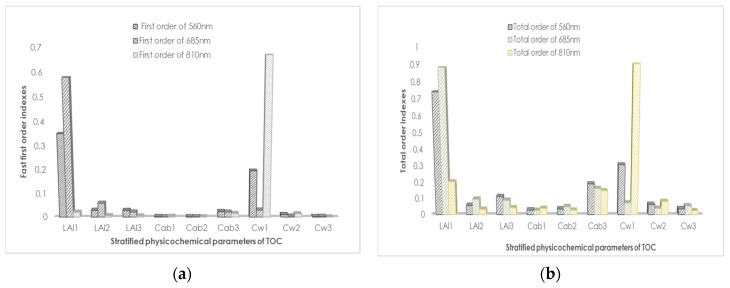
(**a**) Global SA and (**b**) first-order SA of stratified physical and chemical parameters of green (560 nm), red (685 nm), and near-infrared (810 nm) wavelengths.

**Table 1 sensors-20-04570-t001:** Synthetic input parameter data of vertical leaf area index (LAI, $m^{2}\mathrm{leaf}/m^{2}\mathrm{soil}$), chlorophyll ($Chl_{a+b}$, $\mu g/{\mathrm{cm}}^{2}$), and vegetation water content ($C_{w}$, $g/{\mathrm{cm}}^{2}\;\mathrm{fresh}\;\mathrm{weight}$) profile in three-layered canopy scenarios.

Value	Parameter	Top Layer		Middle Layer		Lower Layer	
		Min	Max	Min	Max	Min	Max
Layered value	LAI	0.15	1.2	0.25	2	0.1	0.8
	$Chl_{a+b}$	7.5	82.5	10	110	5	55
	$C_{w}$	0.001	0.07	0.0012	0.07	0.0005	0.04
Standard value	LAI	0.75		1.25		0.05	
	$Chl_{a+b}$	80		60		40	
	$C_{w}$	0.02		0.021		0.01	

**Table 2 sensors-20-04570-t002:** Main input parameters of the SCOPE model and partial parameters of the mSCOPE model.

Parameter	Explanation	Unit	**Standard Valve**	**Range**
$C_{ab}$	Chlorophyll a + b content	$g/{\mathrm{cm}}^{2}$	40	0–100
$C_{dm}$	Leaf mass per unit area	$g/{\mathrm{cm}}^{2}$	0.01	0–0.02
$C_{w}$	Equivalent water thickness	cm	0.015	0–0.05
$C_{s}$	Senescence material (brown pigments)	-	0.1	0–1
$C_{ca}$	Carotenoid content	$g/{\mathrm{cm}}^{2}$	10	0–30
N	Leaf structure parameter	-	1.5	1–3
LAI	Leaf area index	-	3	0–6
${\mathrm{LIDF}}_{a}$	Leaf inclination function parameter a	-	−0.35	−1–1
${\mathrm{LIDF}}_{b}$	Leaf inclination function parameter b	-	−0.15	−1–1
$\varepsilon_{1}$	fluorescence efficiency of photosystem I	-	0.004	0–0.01
$\varepsilon_{2}$	fluorescence efficiency of photosystem II	-	0.02	0–0.05
$\theta_{s}$	sun zenith angle	°	45	0–90
φ	relative azimuthal angle	°	0	0–360
PAR	photosynthetically active radiation	$\mathrm{mol}/m^{2}\cdot s$	1200	0–2200

**Table 3 sensors-20-04570-t003:** Extra parameters of the mSCOPE model.

	mSCOPE				SCOPE
Layer index	1	2	…	N	
Leaf properties	v(1)	v(2)	…	v(N)	$v_{canopy}$
LAI	L(1)	L(2)	…	L(N)	$L_{Canopy}$

Note: Leaf attribute parameters include $C_{ab}$, $C_{dm}$, $C_{w}$, $C_{s}$, $C_{ca}$, and N.

**Table 4 sensors-20-04570-t004:** Vegetation indexes selected in the experiment. Units of the selected parameters are as follows: leaf area index (LAI, $m^{2}\mathrm{leaf}/m^{2}\mathrm{soil}$), chlorophyll ($Chl_{a+b}$, $\mu g/{\mathrm{cm}}^{2}$), and water content ($C_{w}$, $g/{\mathrm{cm}}^{2}\;\mathrm{fresh}\;\mathrm{weight}$).

Vegetation Index	Formula	Related Canopy Parameters
Normalized difference vegetation index (NDVI)	$\left(R_{810}-R_{685}\right)/\left(R_{810}+R_{685}\right)$	LAI [21]
Green chlorophyll index ($CI_{green}$)	$(R_{810}/R_{560})-1$	LAI and $Chl_{a+b}$ [16,35]
Green NDVI ($NDVI_{green}$)	$\left(R_{810}-R_{560}\right)/\left(R_{810}+R_{560}\right)$	$Chl_{a+b}$
Water index (WI)	$R_{900}/R_{970}$	$C_{w}$ [36]
Water ratio vegetation index (WRVI)	$R_{1450}/R_{1940}$	$C_{w}$ [23]
	$R_{1600}/R_{820}$	$C_{w}$ [24,25]
Normalized difference water index (NDWI)	$\left(R_{860}-R_{1240}\right)/\left(R_{860}+R_{1240}\right)$	$C_{w}$ [37]

**Table 5 sensors-20-04570-t005:** Parameter distribution of canopy scenes with different layers. Values in a row, from left to right, represent input values for stratification parameters from top of the canopy to the bottom.

Layer	$\mathrm{LAI},\;m^{2}\mathrm{leaf}/m^{2}\mathrm{soil}$							$Chl_{a+b}$ $,\;\mu g/cm^{2}$						
1	4.78							70						
2	2.1	2.68							50					
3	1.5	2	1.28							40				
4	1.29	1.32	1.22	0.95							30			
5	1.2	0.9	1.5	0.68	0.5							20		
6	0.7	1.02	0.9	0.76	0.8	0.6							10	
7	0.52	0.62	0.7	1.4	0.54	0.54	0.5							5

**Table 6 sensors-20-04570-t006:** Mean, maximum, and minimum value of elected parameter factors for verification.

	Top LAI	Middle LAI	Lower LAI	$\mathrm{Top}\;Chl_{a+b}$	$\mathrm{Middle}\;Chl_{a+b}$	$\mathrm{Lower}\;Chl_{a+b}$	$\mathrm{Top}\;C_{w}$	$\mathrm{Middle}\;C_{w}$	$\mathrm{Lower}\;C_{w}$
max	2.37	1.97	1.05	71.06	76.17	55.05	0.0084	0.0091	0.0094
min	0.24	0.36	0.14	30.08	24.25	15.86	0.0076	0.0080	0.0082
mean	1.48	1.42	0.58	58.95	54.30	37.12	0.0081	0.0086	0.0089

**Table 7 sensors-20-04570-t007:** Upper and lower limits of the selected parameter factors in the sensitivity analysis (SA).

Input Factors	Lower Limit	Upper Limit	Mean Value
LAI1	1	6	3.50
LAI2	0.5	5	2.75
LAI3	0.5	4	2.25
$Chl_{a+b}1$	10	80	45
$Chl_{a+b}2$	10	110	60
$Chl_{a+b}$3	10	60	35
$C_{w}$1	0.001	0.03	0.016
$C_{w}$2	0.001	0.02	0.011
$C_{w}$3	0.001	0.01	0.006

**Table 8 sensors-20-04570-t008:** Running time, error, and efficiency statistics for 7 layers.

Layered Number	Time, Second	Error	Efficiency
1	181.647	0.0037	88.91
2	177.311	0.0051	66.88
3	182.027	0.0026	127.43
4	312.134	0.0018	109.85
5	423.791	0.0015	93.21
6	446.987	0.0045	29.73
7	493.191	0.0101	12.03

**Table 9 sensors-20-04570-t009:** Values of vegetation indexes (VIs), correlation fitting equations between agronomic parameters, and corresponding VIs and the determination coefficient.

Agronomic Parameters	Vegetation Index	Linear Equation	Nonlinear Equation	R^2^
LAI	NDVI	/	y = 0.2222ln(x) + 0.6852	0.9241
	RVI	/	y = 4.259ln(x) + 5.8525	0.9850
	${\mathrm{RVI}}_{f}$	/	y = −0.006ln(x) + 0.0525	0.9707
$Chl_{a+b}$	${\mathrm{CI}}_{\mathrm{green}}$	y = 0.1556x − 0.2441	/	0.9996
	${\mathrm{NDVI}}_{\mathrm{green}}$	/	y = 0.2167ln(x) − 0.0691	0.9917
	${\mathrm{RVI}}_{f}$	/	y = −0.02ln(x) + 0.1267	0.7662
$C_{w}$	WI	y = 3.5046x + 0.9908	/	0.9999
	NDWI	y = 3.5376x − 0.0082	/	0.9931
	$W{\mathrm{RVI}}_{\left(1600\mathrm{nm},\;820\mathrm{nm}\right)}$	y = 8.5485x + 0.1658	/	0.9918
	${\mathrm{WRVI}}_{\left(1450\mathrm{nm},\;1940\mathrm{nm}\right)}$	/	y = −0.146ln(x) − 0.1849	0.9659
	${\mathrm{RVI}}_{f}$	/	y = −0.005ln(x) + 0.0304	0.9884

**Table 10 sensors-20-04570-t010:** Three-tier model-simulated values of VIs, their RMSE, and maximum and minimum values in different experimental groups.

VIs	NDVI	$CI_{\mathrm{green}}$	$NDVI_{\mathrm{green}}$	WI	WRVI1	WRVI2	NDWI
RMSE	0.064	1.28	0.05	0.15	0.48	0.29	0.15
min	0.0013	0.0361	0.0018	0.0113	0.0193	0.0150	0.0180
max	0.255	0.505	0.184	0.193	1.040	0.339	0.191

**Table 11 sensors-20-04570-t011:** First-order sensitivity analysis generated by the Fourier amplitude sensitivity test (FAST) algorithm.

Parameters	Green560	Red685	NRI810
LAI1	0.3507	0.5794	0.0211
LAI2	0.0288	0.0593	0.0058
LAI3	0.0284	0.0213	0.0047
$Chl_{a+b}$1	0.0016	0.0015	0.0028
$Chl_{a+b}$2	0.0012	0.0014	0.0010
$Chl_{a+b}$3	0.0234	0.0212	0.0155
$C_{w}$1	0.1964	0.0298	0.6692
$C_{w}$2	0.0107	0.0042	0.0141
$C_{w}$3	0.0018	0.0031	0.0010

**Table 12 sensors-20-04570-t012:** Total order sensitivity analysis generated by the FAST algorithm.

Parameters	Green560	Red685	NRI810
LAI1	0.7418	0.8846	0.2037
LAI2	0.0567	0.0961	0.0331
LAI3	0.1117	0.0893	0.0425
$Chl_{a+b}$1	0.0277	0.0257	0.0388
$Chl_{a+b}$2	0.0346	0.0490	0.0275
$Chl_{a+b}$3	0.1900	0.1619	0.1480
$C_{w}$1	0.3074	0.0730	0.9058
$C_{w}$2	0.0626	0.0397	0.0821
$C_{w}$3	0.0357	0.0546	0.0238

Note: The number i of LAIi, $Chl_{a+b}i$ and $C_{w}i$ in Figure 11 and Figure 12, Table 11 and Table 12 represent the upper, middle, and lower layers of the canopy respectively consistent with the naming conventions in Table 7.

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
