# Peer review of "Simulation and Verification of Vertical Heterogeneity Spectral Response of Winter Wheat Based on the mSCOPE Model"

_sensors, 2020, doi:10.3390/s20164570_

Round 1
Reviewer 1 Report
This paper evaluates vertical response of layered reflectance, sun-induced florescence chlorophyll and several vegetation indexes of winter wheat and explore the executable ability of mSCOPE. Although it has certain innovation, these many problems in the paper especially in thesis structure, data analysis, important information transition and real-scene verification incompleteness. However, it is suggested that the paper could be accepted after major revisions. The question are as follows:
- The manuscript still needs to be proofread to improve quality of English. Serious problems include sentence fragment, errors in tenses and clauses.
- Introduction and conclusions should be enhanced to explain "why and how" . Plentiful unclear and ambiguous sentences without fundation or the detail data support. For example, “The sensitivity analysis of this study was only applicable to winter wheat and the parameters studied; Global sensitivity analysis can extract the relatively important factors and prevent imperceptible avoidance of factors that are not sensitive to a single variable analysis, such as chlorophyll content in the bottom layer in this study”.
- The thesis writing normative needs be improved.For example the abbreviation should be marked with full name when it is first appearance. Lots of important parameters was not clearly illustrated and some critical indexes and had seldom been mentioned.
- It is unnecessary to waste lots of paper space to repeat the accuracy evaluation index and formula of RMSE and Bias which have the same meaning no matter wherever in real scenes and simulation results.
- The material and methods should include all of methods and processing steps. so that the processing logical and steps is so confused for your disscussion points. For example how many kinds of factors and why they need be disscused.
- The contents of this paper merely simulate vertical variations of different parameters input into the 2D mSCOPE and the verification in real scenes is insufficient and need to be deeply explored.
- Main content or the researches are disscussed on the similation of mSCOPE. However the test or real data collected in the field is limited. In the experiment, 12 different scenarios from dense to sparse without removing ear of winter wheat in booting stage were selected for simulation. also only 12 sets of real data were used to campare and verified the results. more samples should be involved in the research.
- The conclusion merely presents some trivial results and it need to modified logically to correspond the introduction section and the real results and discussions.
Author Response
Response to Reviewer 1 Comments
Thank you very much for taking time to comment on our paper. Your Suggestions helped me a lot to improve this thesis. According to your questions, I had made corrections in the attachment ("(Marked1)sensors-881880.word ") where the red and delete lines were used to mark the modification and deletion places. Please check. If there is still something wrong, your further suggestions are welcome.
Note: The red font represents my reply and the black font is your suggestion.
This paper evaluates vertical response of layered reflectance, sun-induced florescence chlorophyll and several vegetation indexes of winter wheat and explore the executable ability of mSCOPE. Although it has certain innovation, these many problems in the paper especially in thesis structure, data analysis, important information transition and real-scene verification incompleteness. However, it is suggested that the paper could be accepted after major revisions. The question are as follows:
- The manuscript still needs to be proofread to improve quality of English. Serious problems include sentence fragment, errors in tenses and clauses.
Response 1:The syntax and content had been modified as follows.
- Abstract:
Lines 15-28 have been modified.
- Introduction:
added lines 124-132.
Lines 133-146 were deleted and replaced with lines 146-159.
- Materials and Methods:
Deleted lines 162-164, 180-190, 223, 238, 243, 256, 259, 285-330, 357-380 (Content reordering, typesetting, and revision).
Added or modified lines 165-176, 224-229, 244-245, 257-250, 257-258, 259-268, 271, 273, 276, 281(the additional experimental samples are highlighted in red), 353-355, 356.
4. Results
Lines 382, 397, 432-464, 498, 536-549, 554-564, 613-623, 626-636, 654-662, 670-671, 684-688 were deleted.
Lines 383, 398, 403-429, 465, 467-468, 476-477, 494, 498-500, 513, 517-535, 550, 565, 569, 589, 591-612, 623-625, 636-654, 672, 674-681, 688-690, 710-711 have been added or modified.
5. Discussion
Corrected syntax for lines 715 717 718 720 721 722 723 726 727 728.
Added lines 731-738.
Deleted line 739.
6. Conclusions
Deleted lines 773-775, 803-806.
Added lines 772-773、793-801.
- Introduction and conclusions should be enhanced to explain "why and how" . Plentiful unclear and ambiguous sentences without fundation or the detail data support. For example, “The sensitivity analysis of this study was only applicable to winter wheat and the parameters studied; Global sensitivity analysis can extract the relatively important factors and prevent imperceptible avoidance of factors that are not sensitive to a single variable analysis, such as chlorophyll content in the bottom layer in this study”.
Response 2:Ambiguous sentences had been deleted. The necessity to use SA for different crops in different regions when using mSCOPE model were explained.
Lines 674-681: Due to the limited manpower and material resources, it’s difficult to obtain and calibrate all the parameters involved in the model. To reduce the number of parameters, SA should be carried out according to local crop varieties and its growth conditions. On the one hand, SA should be performed before a model is used because the extraction of major parameters improves the accuracy and efficiency of simulation work; On the other hand, Because the physiological characteristics of different crops (such as LAI and plant height) are generally different, the sensitivity of corresponding parameters should be analyzed again to judge if the parameters need to be localized and regionalized.
- The thesis writing normative needs be improved. For example the abbreviation should be marked with full name when it is first appearance. Lots of important parameters was not clearly illustrated and some critical indexes and had seldom been mentioned.
Response 3:A full name marker for the first abbreviation had been added to the summary and body. Descriptions of important missing parameters have also been added.
-
- Lines 33, 38, 40 and 41 in Summary.
- Lines 119、129、271、409 in Introduction.
- Added explain for parameters: Line 353-355, 476-477, 591.
- It is unnecessary to waste lots of paper space to repeat the accuracy evaluation index and formula of RMSE and Bias which have the same meaning no matter wherever in real scenes and simulation results.
Response 4:Redundant formulas had been removed, the sections had been rearrangement, and redundant contents have been removed and improved, as shown in lines 285-326.
- The material and methods should include all of methods and processing steps. so that the processing logical and steps is so confused for your discussion points. For example, how many kinds of factors and why they need be discussed.
Response 5:Modified the logical order and content of materials and methods and Showed the distribution of selection factors (lines 170-176). Section 2.1 introduced the parameterization of mSCOPE model; Section 2.2 introduced the selection of vegetation index; in Section 2.3, two scenarios were designed, which were used for selecting the optimal number of layers and for accuracy assessment using statistical indicators, respectively; in Section 2.4, SA was applied to verify the accuracy. In this study, the main parameters that affected the spectral response of mSCOPE model were discussed, and only 7 main parameters need to be changed in mSCOPE. They were also explained(Note: Leaf attribute parameters include , , , , , and N.). Carotene is 40 percent of chlorophyll and standard values of N, Cs and dry matter content were selected as inputs. The remaining parameters that can be changed and be sensitive to the spectrum were the stratified LAI, Cw and chlorophyll content selected in the paper. Thus, we discussed these three main parameters to explain the generality of the model in the experiment. In the follow-up studies, the effects of the parameters on winter wheat in a specific scenario can be studied according to needs.
Noted: Cs represented the senescent material fraction;N represented the leaf thickness parameters;Cdm represented the Dry matter content.
- The contents of this paper merely simulate vertical variations of different parameters input into the 2D mSCOPE and the verification in real scenes is insufficient and need to be deeply explored.
Response 6:In addition to the twelve experiments mentioned in the paper, four-stratification nitrogen experiments were conducted. The appendix is an experimental discussion of over nitrogen, nitrogen deficiency, and normal nitrogen levels. This paper didn’t cover the nitrogen study part, so it’s not included in the text. The monitoring of nitrogen level will be further studied in the follow-up study. On the other hand, this paper mainly focused on describing the response of the spectrum to the main input stratification parameters, and coupled with the difficulty of measuring the internal stratification spectrum due to experimental conditions limited. Thus, only the overall distribution of the spectrum of canopy was discussed. Subsequent studies will also focus on the collection of stratification parameters and spectra. And in-depth analysis in practical scenarios such as nutrient diseases will be promoted with emphasis on experimental verification.
- Main content or the researches are disscussed on the simulation of mSCOPE. However the test or real data collected in the field is limited. In the experiment, 12 different scenarios from dense to sparse without removing ear of winter wheat in booting stage were selected for simulation. also only 12 sets of real data were used to campare and verified the results. more samples should be involved in the research.
Response 7: Nine groups of samples were added on the basis of the original experimental data, and the experimental results were analyzed. 21 groups of samples were shown in Figure 1 (line 281). The results after supplementary experiments were shown in line 636-672.
- The conclusion merely presents some trivial results and it need to modified logically to correspond the introduction section and the real results and discussions.
Response 8:As shown in lines 636-672, the logical order of the conclusion was adjusted to match with the order of result. The simulation and validation are discussed respectively. FAST was a separate validation analysis method, so it was explained as a separate part.

Reviewer 2 Report
The manuscript, Simulation and verification of vertical heterogeneity spectral response of winter wheat based on the mSCOPE model, generally well-written and clearly presented.
I don't have major comments for authors.
The only concern I have is about the implications of this study. I think it would be better if authors could talk a little bit on How the findings of this study could be used in Wheat cultivation/productivity improvement etc. in the discussion.
Author Response
Response to Reviewer Comments
Thank you very much for taking time to comment on our paper. Your Suggestions helped me a lot to improve this thesis. According to your questions, I had made corrections in the attachment ("(Revised2) sensors-881880.word ") where the red and delete lines were used to mark the modification and deletion places. Please check. If there is still something wrong, your further suggestions are welcome.
Note: The red font represents my reply and the black font is your suggestion.
Point: The manuscript, Simulation and verification of vertical heterogeneity spectral response of winter wheat based on the mSCOPE model, generally well-written and clearly presented.
I don't have major comments for authors.
The only concern I have is about the implications of this study. I think it would be better if authors could talk a little bit on How the findings of this study could be used in Wheat cultivation/productivity improvement etc. in the discussion.
Response:The spectral differences of winter wheat under different nitrogen stress were discussed, as shown in lines 520-527.
Nitrogen and chlorophyll are closely related. We simply verified that winter wheat with different nitrogen content had a good performance, indicating that this model had a broad application prospect. This conclusion can also be extended to other fields where practical scenarios are combined. For example, diseases and insect pests may cause yellower leaves in the middle layer. Common spectral observation method generally focuses on the top of the canopy. However, this way may cause trouble. When the top layer of leaves can detect spectral changes deterioration, the disease has intensified. Using mSCOPE model to invert physiological indexes such as VIs of canopy interior stratification is undoubtedly a good way to solve the difficulty of internal spectral measurement.
The appendix ("(Revised2) sensors-881880.word ") was an experimental discussion of excessive nitrogen, nitrogen deficiency and normal nitrogen levels. Nitrogen was not involved in this paper, so it was not included in the text and it is just for a reference. The monitoring of nitrogen levels will be further studied in future studies.

Round 2
Reviewer 1 Report
This paper evaluates vertical response of layered reflectance, sun-induced florescence chlorophyll and several vegetation indexes of winter wheat and explore the executable ability of mSCOPE.
It has certain innovation, however the most problems in the paper especially in real-data verification incompleteness. The plant samples are limited, so that the robust of simulation model and the influences of natural measurement is difficult to analyze. However, it is suggested that the paper could be accepted after revising. The question are as follows:
1\ How to conduct the field experiment to prove the simulation results, especially in the layer classification?
2\ Why you select such vegetation indices? the result of table 9 shows that the RVI is good for all of agronomic parameters. more disscution should be presented.
3\ Sensitivity analysis of canopy reflectance should be improve to explore the vertical profiles of fied crops.
Author Response
Thank you very much for your further comment on our paper. Your Suggestions helped me a lot to improve this thesis. According to your questions, I had made corrections in the attachment ("minRevised-sensors-881880.word ") where the red and delete lines were used to mark the modification and deletion places. Please check. If there is still something wrong, your further suggestions are welcome.
Note: The red font represents my reply and the black font is your suggestion.
It has certain innovation, however the most problems in the paper especially in real-data verification incompleteness. The plant samples are limited, so that the robust of simulation model and the influences of natural measurement is difficult to analyze. However, it is suggested that the paper could be accepted after revising. The question are as follows:
1\ How to conduct the field experiment to prove the simulation results, especially in the layer classification?
Response1:I added a further description of the hierarchical algorithm (172-191). In order to simplify the calculation and facilitate the implementation of the algorithm, the average height hierarchical way was carried out. The mSCOPE model integrated the vertical changes of vegetation attributes, but didn’t consider the horizontal changes. Therefore, mSCOPE can be regarded as a 2D model. The mSCOPE model maintained the same model structure and output were the same as SCOPE, but adopted different solutions for incident emission radiation transmission in vegetation canopy. The reflectivity was solved by bottom-up "adding" method and the flux profile was calculated by top-down "peeling" method. The hierarchical algorithm steps are as follows:
- The vertical heterogeneous layer was divided into several homogeneous layers
- Starting from the bottom (bottom up) homogeneous layer, the model calculated the surface reflectance of the combined system between bottom layer (such as soil) and the next layer;
- In Step 2, a new uniform vegetation layer was added to the surface of the original system to calculate the surface reflectance of the new system;
- Repeat step 3 until all isomorphism layers were added;
- Once the surface reflectivity of each vertical plane was obtained, the flux profile can be calculated from top to bottom according to the incident flux (known quantity).
In conclusion, stratification can be extended to any multi-stratification mode between 2-60 layers on the premise of considering both accuracy and cost performance. Considering the labor cost and operation difficulty, field measurement verification data generally adopt three layers according to the highly uniform stratification of winter wheat. The parameter values of each layer were obtained through field measurement. Then they were inputted into the input table provided by mSCOPE, and the program will automatically call to simulate the results.
2\ Why you select such vegetation indices? the result of table 9 shows that the RVI is good for all of agronomic parameters. more disscution should be presented.
Response2:The reason why I choose these vegetation indexes is that I have selected several representative ones with good performance in the papers of the predecessors based on the reference materials and suggestions from my tutor. According to the typical characteristic bands, different vegetation indexes were combined to simulate and verify the experiment. The results showed that the effect is good. Discussion of RVI (ratio vegetation index) was described in Line 446-453.
3\ Sensitivity analysis of canopy reflectance should be improved to explore the vertical profiles of fied crops.
Response3: Sensitivity analysis was further discussed and Line 505-514 was added. Figure 11(Diagram of first-order and global SA)was added to explain difference between first-order SA and global SA more intuitively.